# Concepts of a Modular System Architecture for Distributed Robotic Systems

**Uwe Jahn \*, Carsten Wolff and Peter Schulz**

IDiAL Institute, Dortmund University of Applied Science and Arts, 44227 Dortmund, Germany;
carsten.wolff@fh-dortmund.de (C.W.); peter.schulz@fh-dortmund.de (P.S.)
**\*** Correspondence: uwe.jahn@fh-dortmund.de; Tel.: +49-231-9112-9661

**Abstract:** Modern robots often use more than one processing unit to solve the requirements in robotics. Robots are frequently designed in a modular manner to fulfill the possibility to be extended for future tasks. The use of multiple processing units leads to a distributed system within one single robot. Therefore, the system architecture is even more important than in single-computer robots. The presented concept of a modular and distributed system architecture was designed for robotic systems. The architecture is based on the Operator–Controller Module (OCM). This article describes the adaption of the distributed OCM for mobile robots considering the requirements on such robots, including, for example, real-time and safety constraints. The presented architecture splits the system hierarchically into a three-layer structure of controllers and operators. The controllers interact directly with all sensors and actuators within the system. For that reason, hard real-time constraints need to comply. The reflective operator, however, processes the information of the controllers, which can be done by model-based principles using state machines. The cognitive operator is used to optimize the system. The article also shows the exemplary design of the DAEbot, a self-developed robot, and discusses the experience of applying these concepts on this robot.

**Keywords:** robotics; mobile robots; distributed systems; system architectures; operator–controller module (OCM); health monitoring; modular systems; single-board computer (SBC); cloud computing

## 1. Introduction

The design and advancement of complex technical systems is a challenge for developers. A particularly complex system is the robotic system, as robots utilize extensive functions as well as a high number of actuators and sensors. Autonomously acting robots need performant hardware to perform complex software algorithms.

Past robots often used one computer as the processing unit of the systems, implementing software without using software architecture approaches suitable for robotic systems. These software implementations also often lack monitoring features which can be useful due to safety reasons in mobile robots. Using a single processing unit also prohibits the extension of the system for changing system demands in a modular way.

With the emergence of small single-board computers (SBC) as more energy-efficient and more suitable solutions for processing units of a mobile robot, more and more SBCs are used. Multiple SBCs or microcontrollers are combined to a distributed system to obtain more computing power or better dependability. Furthermore, modular mobile robots like the AMiRo (Autonomous Mini Robot) [1] have been released. These modular robots can be extended easily and mostly use multiple processing units.

The modular approach and the use of multiple processing units lead to a distributed system within a single robot. The system architecture of those distributed systems often lacks solutions to fulfill requirements needed in robotics (e.g., like real-time constraints). Often distributed systems

grow historically [2]. First one SBC is used as a processing unit, and later other SBCs for different sensors, actuators, or Human–Machine Interfaces (HMI) are added. Not focusing on the architecture and planning to include new components for future projects lead to problems integrating those new components. Comfort features, like visualization tools, could be mixed with actuators which must fulfill hard real-time constraints in one operating system or framework, for example, in ROS (Robot Operating System [3]). Thus, the architecture of those distributed systems within one robot becomes much more important.

This article describes a new concept for a modular system architecture for robotic systems, focusing on distributed systems within a mobile robot. The article also includes an exemplary implementation of the proposed architecture on a mobile robot, the DAEbot.

This article is organized as follows: Section 2 presents the current state of the art focusing on architectures used in robotics and the Operator–Controller Module (OCM). Section 3 describes the conceptual model, including the architecture, communications, and how requirements on mobile robots have been integrated. Section 4 shows the exemplary implementation of this conceptual model with the DAEbot. Section 5 shows some results of this project. The last section summarizes the article with a conclusion and outlook for upcoming projects.

This article is an extended version of a previous article at the 24th International Conference on Information and Software Technologies (ICIST 2018) [4]. The original conference paper has been extended with a more detailed state of the art and the new central chapter "Conceptual Model".

## 2. State of the Art

### 2.1. Single-Board Computers (SBCs) in Robotics

The usage of SBCs is a step forward to developer-friendly robot designs. In contrast to using industrial computers in robotics, SBCs are used in many areas, dedicated IDEs (integrated development environments) are available, and communities can help within the development process. Thus, developers do not necessarily need expert knowledge of the processing unit's hardware and can start with using available operating systems.

Many design approaches (e.g., of mobile robots) use one single processing unit (e.g., SBC) to control the complex technical system. A (single) Arduino Uno is used to run a surveillance robot for outdoor security [5], a microcontroller controls an underwater robot [6], and a mini PC is used to operate a hexapod robot for subsea operations [7].

### 2.2. Condition Monitoring

The monitoring of the condition of a technical system can be used to detect anomalies and off-nominal operating conditions [8]. To monitor the condition of hardware components, for example, electrical motors help to detect problems earlier to prevent any hardware damage and to process a fault diagnosis with the monitored data [9]. Strategies can be planned to react to failures (e.g., to trigger a repair process [10]).

### 2.3. Architectures for Robotic Systems

Robotic systems are often based on an ROS (Robot Operating System), which is widely accepted as the standard framework for robotic system development [11]. ROS helps to develop a robot easily by adding many ROS packages to a software stack, but developers are not led to focus on the robot's component structure or the underlying architecture. ROS can be implemented on various operating systems (e.g., on real-time operating systems (RT-OS) or non-real-time operating systems). This increases the chance to mix hard real-time and "soft" real-time using ROS packages with different underlying operating systems. ROS itself is one example for a modular system, as different ROS packages can be combined in a modular manner.

The research on modular architectures is a well-established research topic (e.g., presented in [12]).

In 1998, Mark W. Maier [2] published architecting principles for systems-of-systems. His work defines systems-of-systems and describes architectural principles and the different layers of communications.

In [13], among other things, a design structure matrix is presented to analyze the modularity and complexity of system-of-systems (SoS). It also describes different architectural designs, like a centralized architecture, a distributed estimation architecture, and a distributed tasking architecture.

The literature research also shows many architecture approaches for robotic systems. In [14], a systematic mapping study on software architectures for robotic systems is described. The brief comparison and quantification regarding a quality score were used to take a more in-depth look into some of the presented architectures and extend the list with their own literature research results.

Corke et al. [15] present DDX (Dynamic Data eXchange) as distributed software architecture for robotic systems. This architecture divides the system into clients and computers. DDX provides two programs to share data within the clients (store) and with other computers (catalog). In contrast to our approach, DDX does not structure the architecture hierarchically.

In [16], an architecture for distributed robotic systems is outlined, which divides the robot into a client and server part. One disadvantage of this approach is that the server part of this architecture is necessary and not optional to run features like state machine execution.

Another software architecture was developed using the Orca [17] framework. The presented architecture focuses on the abstraction layer of a user interface using a base station with a user interface, a ground station, and two operators to interact with the ground vehicle. Lower abstractions, like the internals of the used components, are not included in the ORCA framework.

Yang et al. [18] outline a high-level software architecture. The robot's software architecture is distributed into a reactive layer, where sensing and acting with the environment is done on an adaptive layer. The adaptive layer includes modeling, planning, and monitoring of a mobile robot. Both layers are located on one computer, not on different processing units as a distributed system.

For the SERA (Self-adaptive dEcentralized Robotic Architecture) approach [19], related architectures have been reviewed. The SERA approach consists of a distributed system of robots and a central station. SERA uses three layers on the robots, the component control layer, change management layer, and the mission management layer, which are based on Kramer et al. [20]. In SERA, the three layers on the robot are located on one processing unit, not on different units as a distributed system on the robot itself. A goal for the SERA is a decentralized architecture for collaborative robots with the central station.

## 2.4. Operator–Controller Module (OCM)

In 1979, Ropohl [21] introduced a hierarchical multilayer structure for a general system, which can be a technical system (see Figure 1).

The lowest layer, the so-called execution system, interacts directly with sensors and actuators, the physical world of the technical system. The execution system itself interacts with the material, the physical part of this Cyber-Physical System (CPS).

The information system receives and transmits information from and towards the execution system. This information is used for the processing of the sensor and actuator data from the physical world.

The top layer of this three-layer-approach is used for setting the target(s) of the activity system. This layer is arranged hierarchically above the other two layers.

The explicit distribution of one system into a hierarchically organized structure is also used with the Operator–Controller Module (OCM).

The OCM appeared in Naumann's 2000 dissertation [22] as a concept for intelligent mechatronic systems. Following Ropohl's approach, the OCM decouples the controller for the direct interaction with the physical system and its sensor and actuator signals from a reflective operator and a cognitive operator (see Figure 2). In [23], the three layers are described in detail.

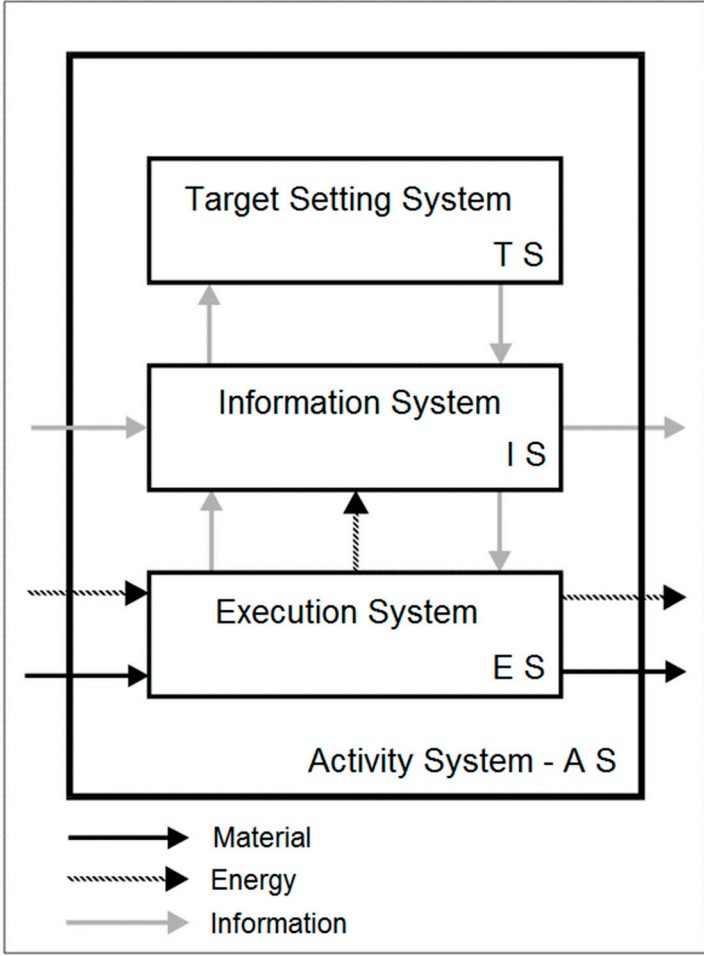

**Figure 1.** Ropohl's Activity System [21] (Translated from German).

The controller level stands for the motor loop with direct access to the technical system. The controller(s) are located at the action level where time-discrete constraints need to be fulfilled. Interacting with the physical world leads to the need to guarantee compliance with hard real-time constraints. The controller level can utilize many controllers, each having different configurations (e.g., emergency mode or standby).

The reflective operator does not have direct access to the technical system. The reflective operator is used to supervise the controllers in the reflective loop by processing the information from the controllers. This information is processed event-driven. The reflective operator needs to decide which configurations of the controllers are used based on the robot's environment and its task. Also, the reflective operator serves as a middleware between the controller and the cognitive operator and decouples hard real-time parts of the system from soft real-time parts.

The cognitive operator is the highest level in this architecture. In a cognitive loop, the cognitive operator is used to improve the behavior of the system on its planning level. This level includes strategies for self-optimization.

The concept of the OCM was initially developed for self-optimizing mechatronic systems [22]. The OCM was refined later and was used in several projects of the research cluster "Self-optimizing systems in mechanical engineering (SFB614)" [24] (e.g., the RailCab project [25]). Metropolitan energy systems [26] and ORC turbines [27] have also used the OCM design.

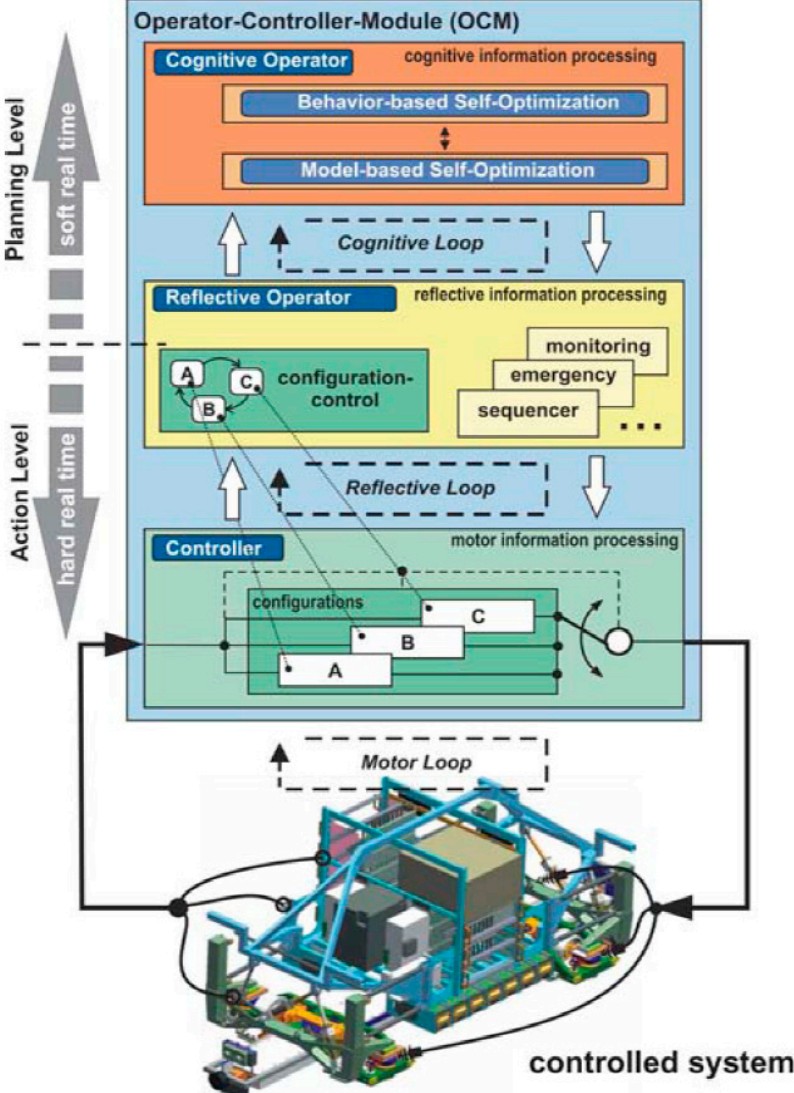

**Figure 2.** Architecture of the Operator–Controller Module (OCM) [23].

## 3. Conceptual Model

The conceptual model of a modular software architecture for distributed robotic systems is based on the presented OCM. The conceptual model describes the implemented adaption of the OCM approach on the field of robotics to fulfill requirements on such a robotic system.

### 3.1. Architecture

The proposed architecture consists of the three OCM layers: the controller(s), reflective operator, and cognitive operator (see Figure 3). The controllers interact directly with actuators and sensors in a motor loop. The reflective operator processes the data from the controllers in an operating loop. The reflective operator decides in which configuration the controllers are running and controls the robots to fulfill their given task(s) (e.g., to navigate through a room). The highest layer, the cognitive operator, runs an optimizing loop for the lower layers to optimize the behavior of the system (e.g., to reduce power consumption). A Human–Machine Interface (HMI) can be implemented to both of the two operators, as an HMI is not required to fulfill hard real-time constraints. The architecture is divided into two parts: the local part, which can run the robot autonomously, and the cloud area, to add additional computing power (e.g., to calculate complex (optimizing) software algorithms).

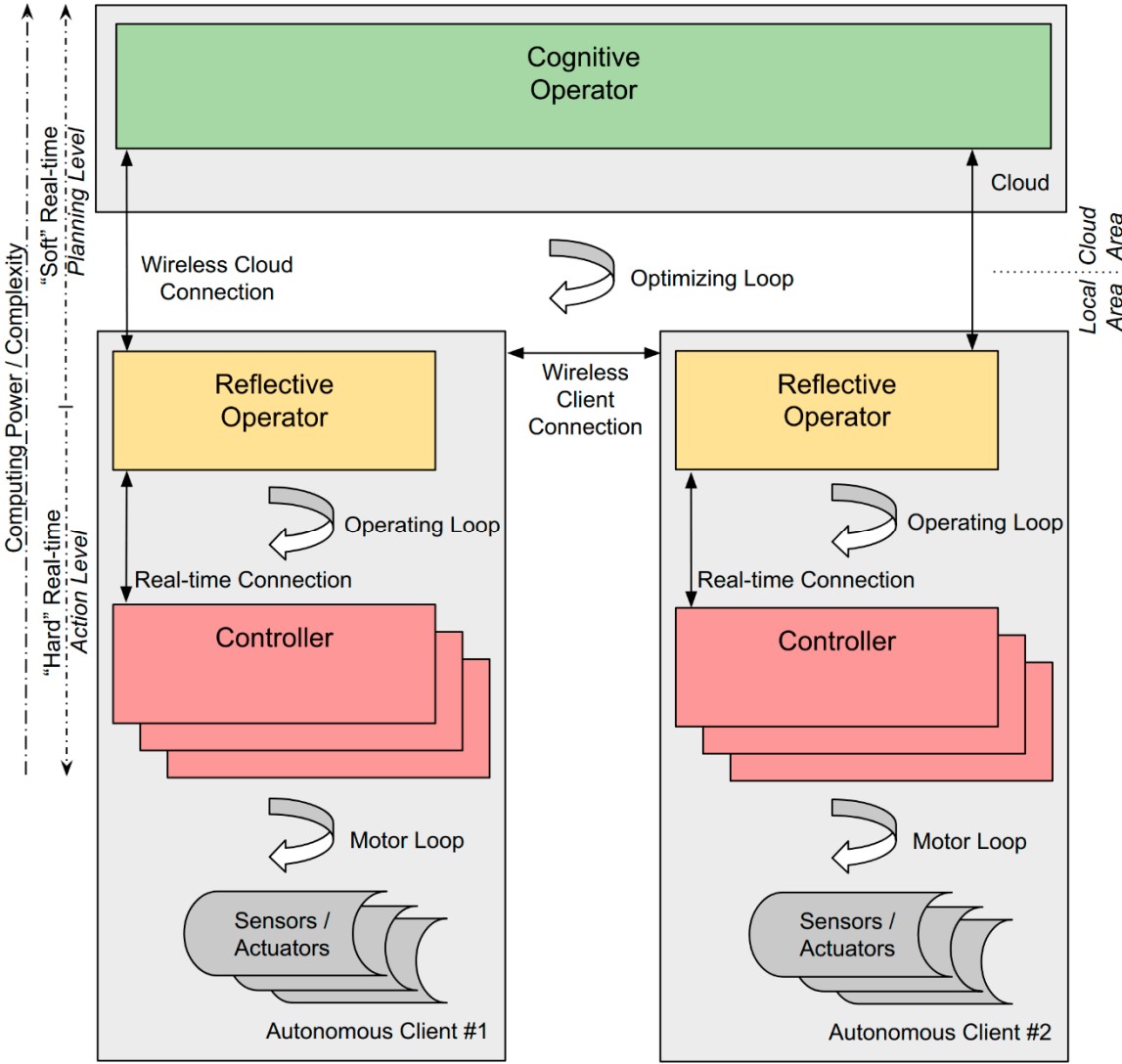

**Figure 3.** Conceptual model for an architecture for a distributed robotic system.

## 3.2. *Autonomy* vs. *Cloud Computing*

The article "Robots with their heads in the clouds" [28] argues that "cloud computing could make robots smaller and smarter". Modern mobile robots need to combine the ability to compute a high amount of sensors, actuators, and software algorithms with the restriction of limited space for powerful processing units and the needed battery packs to run those processing units. The proposed approach divides the robotic system into two areas: one locally on the robot and one in the suggested cloud.

The local area, consisting of the controllers and the reflective operator, is set up as an autonomous client. Even without the connection to the cognitive operator, the robot needs to interact with its environment and to fulfill its tasks. Unlike the original OCM approach, our architecture implements the cognitive operator as a cloud service. The benefit of this distribution is that the local part can be equipped with processing units which are low on energy but also do not have the computing power to solve complex software algorithms (e.g., particle filters with a high amount of particles, or neural network applications [29]). The cloud part allows adding this computing power while not using the limited battery power of the mobile robot. A real-time connection to the cloud-based cognitive operator is challenging [30] and not necessary, as the reflective operator defines the immediate reaction of the robot. This ideally fits into the Internet of Things (IoT) domain and opens the path for

seamless integration of technical systems into virtual environments, leading to real Cyber–Physical Systems (CPS).

### 3.3. Time-Discrete vs Event-Driven

One advantage of the OCM is the division into time-discrete and discrete logic (or event-driven) behavior [31]. As in mechatronic systems, hard real-time constraints also need to be fulfilled in robotics. The OCM divides the system into an action level (time-discrete with hard real-time constraints) and a planning level (event-driven with soft real-time constraints).

The controllers are mostly based on time-discrete constraints. That means that the data or algorithms processed by the controllers need to be completed within a defined "hard" real time, without exceptions. The controllers are directly interacting (action level) with the system's sensors and actuators and therefore need to guarantee compliance with hard real-time constraints. In case of an emergency, the controller, for example, needs to stop the system's motors without any delays (e.g., fail-safe).

The reflective operator can be implemented with lower real-time constraints (soft real-time). Information processing depends on the data from the controllers and is, therefore, using event-driven behavior such as event-triggered function calls. The reflective operator also implements the interface to the planning level with the cloud-based cognitive operator. Comfort features, like visualization tools or HMI interfaces, are only allowed to be used on the planning level. Processing those comfort features could create latencies, which should never interrupt hard real-time processes.

This isolation between both levels shall ensure that hard real-time tasks will never be harmed by lower-prioritized tasks.

### 3.4. Communications

The communications in the architecture are designed to serve the hard and soft real-time conditions. On the action level (see Figure 3), the communication should be done in real-time, and on the planning level (see Figure 3), a soft real-time connection is sufficient.

Furthermore, the communication on the modular robot (local area) should be done in a bus network topology and not with a point-to-point connection. This makes it easy to add new components to the system, without changing the network topology [12]. Another advantage of a bus is that watchdogs which are listening to the whole (local) robot agent's communication for analysis or error-detection features can be added. A widely used network bus in automotive and robot application is the CAN-Bus. CAN has low hardware requirements and is also available in a real-time version [32].

It is useful to keep the number of CAN frames low to increase the possibility of colliding CAN frames (e.g., by adapting the publisher rate of a sensor depending on its need for the system).

Figure 4 describes this adaptive publisher frequency with the example of ultrasonic sensors attached on all four sides of a mobile robot. Data from the front ultrasonic sensor is essential if the robot is driving forward. The front ultrasonic sensor will publish its data with 20 Hz. If the robot stands still, all four sensors are running in a standby mode, publishing its data only with 1 Hz.

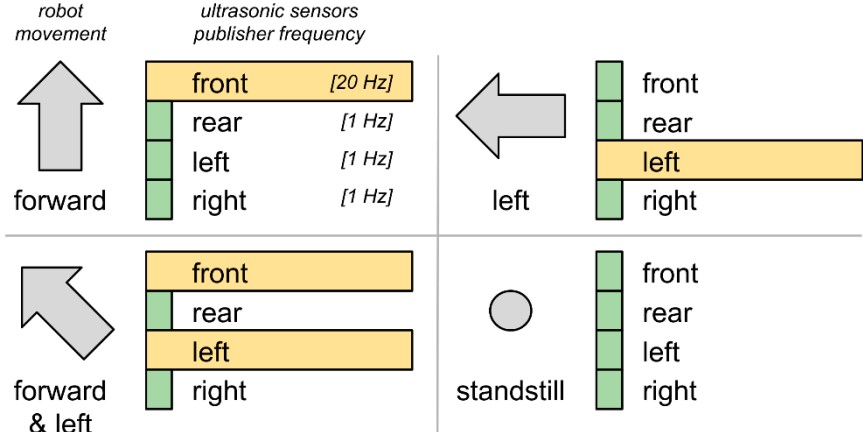

**Figure 4.** Adaptive publisher frequency (example: ultrasonic sensors).

The cloud connection is typically a Machine-to-Machine (M2M) communication architecture [30].

### 3.5. Modularity

In [12], the importance of the concept of modularity, especially in modern robotics, is described. Robots need to adapt to future tasks or new environments (e.g., by connection of a new sensor to recognize objects) to be future-proof. These new sensors can easily be included in our architecture (e.g., by adding another controller to the robot). The distribution of the processing units to different components within one robot leads into a modular design approach. New components only need to be included in the system's communication bus system.

### 3.6. Computing Power and Complexity

The complexity of a robot's software and the high amount of sensor and actuators can easily be adapted to the OCM architecture using multiple processing units within one robot for the controller(s) and the reflective operator. One advantage of single-board computers (SBCs) or microcontrollers in mobile robots is the device size and energy efficiency. With the usage of multiple processing units in a single robot, the computing power can be adapted to the needed complexity (e.g., adding another controller to interact with new hardware sensors or actuators).

Mostly, the complexity increases from the bottom of the three-layer architecture to the top (see Figure 3). On the lowest level, the controllers preprocess sensors and actuators in control loops. Those processing tasks are simple and must fulfill hard real-time constraints. On the next layer of the processing hierarchy, the reflective operator runs more complex software algorithms (e.g., mapping or computer vision algorithms). Very complex or data-intensive algorithms are running on the cloud-based cognitive operator. This allows computer power to be added to or removed from the mobile robot as it is needed. The cloud-based algorithms can also be used when the robot is disconnected (e.g., to simulate different optimizing and learning strategies).

Also, the modular approach makes it possible to add the most suitable hardware for tasks. For example, an FPGA board can be added to the robot to run computer vision algorithms much more efficiently than ARM-based processing units.

### 3.7. Safety and Redundancy

The dependency on a single processing unit is one disadvantage of such single-processing unit systems. If the only processing unit fails, the system itself will fail. With complex networks, a technical system's robustness can be increased (e.g., by adding redundancies [33]). Using a distributed system in one robot has the advantage of having such a network of many processing units onboard which can bring the system into a save state or even take over tasks of the failing processing unit.

It is recommended to implement watchdogs to check the conditions of the robot to detect failures of the system. This can either be done by adding another SBC, as a watchdog listening to the communication bus, or each SBC can run global and local watchdogs. A local watchdog can be used to prevent misbehavior of a single component (e.g., running the robot's wheels even if the same component detects an obstacle). Global watchdogs can be used to detect communication loss (e.g., running the robot's wheels even if the connection to the reflective operator is lost).

The modular design approach also supports the use of backup or redundant components. Backup components can be added to the system to take over tasks if a component is struggling.

### 3.8. Adaptive Component and Sensor Usage

Another challenge for mobile robots is to be as energy-efficient as possible. The proposed approach uses mostly energy-efficient SBCs or microcontrollers, but due to the number of components, the overall energy consumption is high. To prevent this, the robot should be able to shut down or start components and sensors as they are needed. If, for example, the usage of a camera is unnecessary, the component can be switched off completely. It should also be possible to turn sensors and actuators with high energy consumption off and on.

## 4. Design of the DAEbot

The DAEbot (Distributed Architecture Evaluation Robot) has been developed to evaluate distributed architectures within one robot, in this case, the presented conceptual model based on the OCM. The DAEbot (see Figure 5) consists of a modular structure, which can be extended with actuators and sensors for different applications (e.g., with a depth camera to navigate autonomously to a given location). The cognitive operator for the DAEbot is under development.

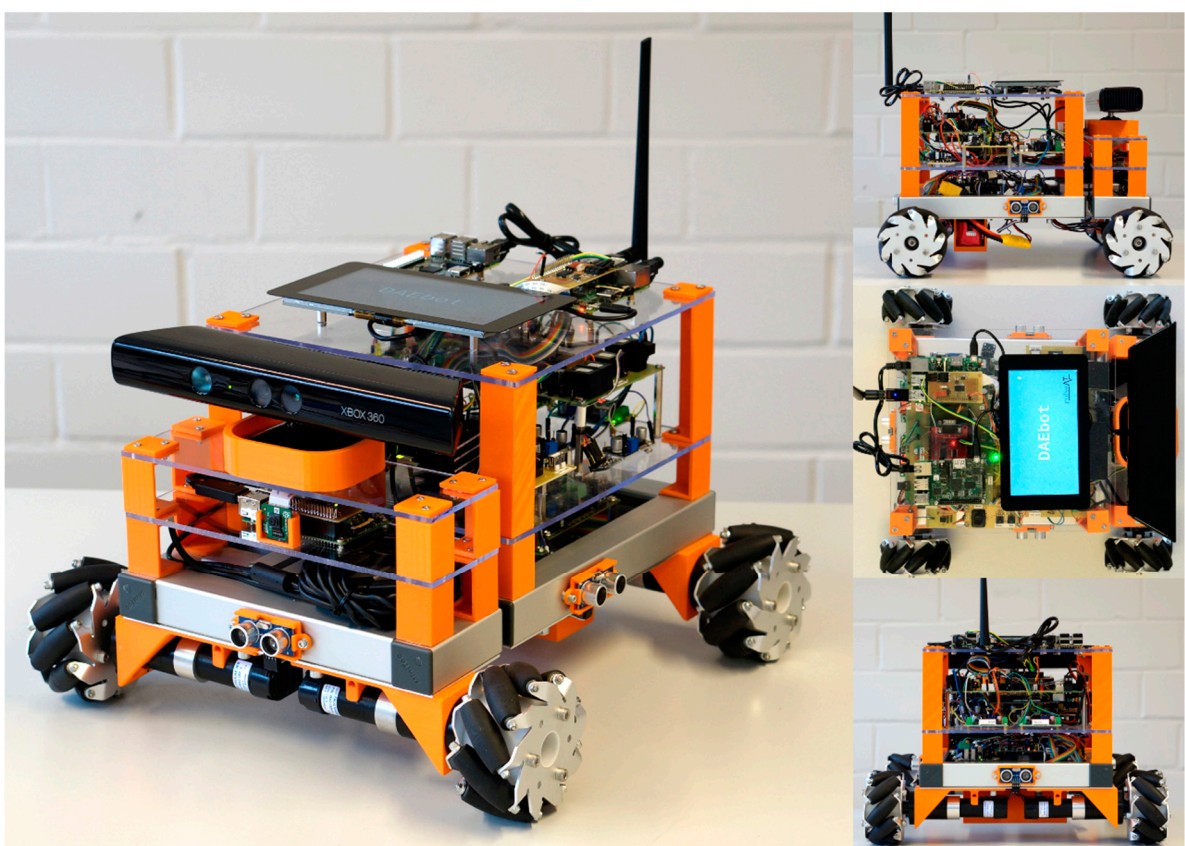

**Figure 5.** DAEbot as a demonstrator for the operator–controller-based distributed system.

All microcontrollers of the DAEbot are SBCs. The robot basis is a self-designed platform which is driven by four mecanum wheels [34]. The robot basis contains motor encoders, motor drivers, and ultrasonic sensors. This chapter describes all components of the DAEbot, including the implemented communication framework and safety and redundancy functionalities.

### 4.1. Communication

The primary communication in the DAEbot is done via Controller Area Network (CAN) in a publisher–subscriber concept. The de facto standard in robotics ROS (https://wiki.ros.org/ROS/Concepts) is also based on this publisher–subscriber concept. A publisher (e.g., an ultrasonic sensor) offers its data to the system. A subscriber (e.g., a processing algorithm) can subscribe to this sensor data for its purpose.

For this concept, a self-developed CAN ID framework has been implemented (see Figure 6), which allows changing the priority of messages dynamically. Therefore, the common CAN identifier (11 bits) is altered into three parts: the C/S bit, two priority bits, and an eight-bit identifier. The first bit defines if the attached CAN frame contains sensor data from a controller (C/S = 1) or command data from the operator (C/S = 0). With the next two bits, the priority of the CAN frame can be configured as urgent (00), high (01), medium (10), or low (11). As these priorities can be reconfigured for each message, critical messages can be prioritized (e.g., the priority of the message to stop the robot's wheels can be set to urgent if an emergency stop needs to be performed). Low identifiers are prioritized over higher ones in CAN on the hardware site, meaning that command data is always handled prior to sensor data, following its personal priority. The last eight bits contain a personal identity for each actuator and sensor value of the system. As a result, a CAN frame can be transmitted to the same entity either to switch to a different mode by changing the C/S bit or to transmit the sensor data.

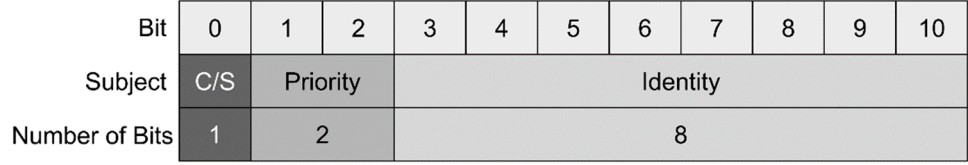

**Figure 6.** CAN communications ID for the DAEbot.

The DAEbot communications framework uses three modes. Mode "0" is used to switch off a publisher (e.g., if the sensor data is not needed for the current task of the robot). With a mode "1" message, a publisher can be (re)configured by attributing the sensor sample time. With a mode "2" message, the operator asks the controller for one-time data transmission.

The publisher–subscriber implementation of the DAEbot can be configured to the exact needs of the robot's application in order to use the controller's hardware resources efficiently. The use of the priority bits in the CAN ID also helps to prioritize critical over less-important information (e.g., to prefer real-time controller messages over non-time-critical command messages).

In addition to the local CAN communication, the DAEbot is connected via WiFi to the cloud-based cognitive operator.

### 4.2. Modular Design

The DAEbot is based on SBCs which are connected via CAN. This modular approach makes the DAEbot future-proof (e.g., by upgrading the used SBC with new ones). More SBCs can be added to the system for future tasks (e.g., to add more efficient or more powerful processing units to the system).

### 4.3. Components

The DAEbot utilizes five SBCs as operator or controller units implementing the OCM approach. The three controllers (see Figure 7, red bottom layer) are directly connected with several actuators

and sensors (grey). The reflective operator and the reflective operator+ (yellow middle layer) are interacting with the controllers via CAN. The cognitive operator (green top layer) will be used to optimize the distributed system.

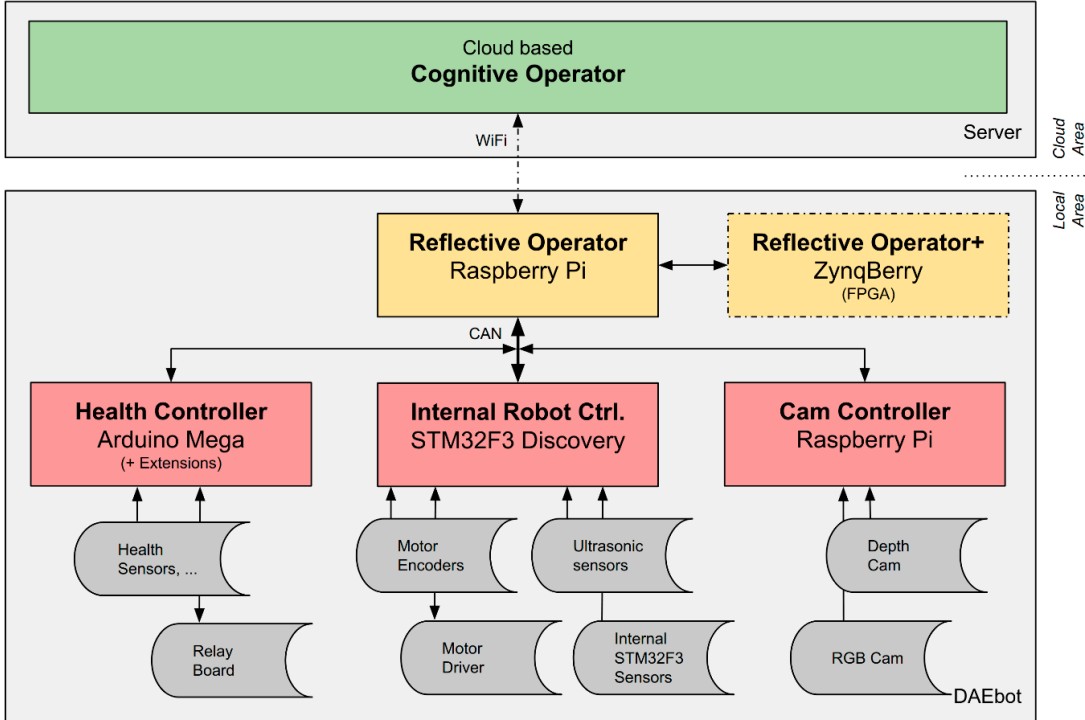

**Figure 7.** Components of the DAEbot.

The mechanical design of the DAEbot itself is designed with spatial layers for the controllers and a layer for the local operators (see Figure 1) to demonstrate the software architecture. As the presented architecture is modular, more components can easily extend the system for future tasks. Different SBC/microcontrollers are used to evaluate those popular processing unit families and to demonstrate the suitability of them in such a mobile robot. The DAEbot combines multicore SBCs (Raspberry Pi (https://www.raspberrypi.org/products/)), an Arduino Mega (https://store.arduino.cc/arduino-mega-2560-rev3) with lots of GPIOs, an STM32F3Discovery (https://www.st.com/en/evaluation-tools/stm32f3discovery.html), and an FPGA board (ZynqBerry TE0726-03M (https://shop.trenz-electronic.de/en/TE0726-03M-ZynqBerry-Zynq-7010-in-Raspberry-Pi-form-factor)). A (Raspberry Pi) touch display is used for visualization features.

### 4.3.1. Internal Robot Controller

The Internal Robot Controller (IRC) connects and processes the actuators and sensors which are necessary for the movement of the robot. The STM32F3Discovery has been extended with a self-developed board to add a CAN interface, pinouts, and additional ports and interfaces to connect the periphery. The IRC runs the motor drivers and receives rotation data from the motor encoders. It also connects four ultrasonic sensors and processes a magnetometer and a gyroscope.

The microcontroller uses the operating system FreeRTOS [35] to satisfy the real-time requirements. The IRC implements safety features like interrupting the motors if the ultrasonic sensors recognize an obstacle movement direction.

### 4.3.2. Cam Controller

The Cam(era) Controller (CC) extends the DAEbot with image sensors, which are used for an autonomous navigation application to demonstrate the robot's capabilities. First, a Raspberry

Pi RGB camera is connected to the Raspberry-Pi-based CC. Secondly, a Microsoft Kinect (https://developer.microsoft.com/en-us/windows/kinect) depth camera is set up. Both cameras use OpenCV (https://opencv.org/) to recognize objects or obstacles. The Raspberry Pi RGB camera has the advantage of low power consumption compared to the Kinect. The Kinect outperforms the RGB camera in a low-light environment or darkness. The Kinect is also used to get depth data from objects. The operator's task is to decide which camera is best for the current situation and task (e.g., in case of darkness, the depth camera would be the obvious choice over the RGB camera). The other camera can be disabled or, in the case of the Kinect, be switched off entirely with a relay. A CAN shield is used to include the CC into the local DAEbot network. A custom Linux-based RTOS has been built with the Yocto project (https://www.yoctoproject.org/).

### 4.3.3. Health Controller

The Health Controller (HC) is used to collect the condition data of the robotic system. It provides different types of information about the system. This information can be used by the operator(s) to (re)configure the system behavior (e.g., in case of critical battery or abnormally high motor temperatures).

First, the HC collects information about the energy of the systems. For this, the current power consumption of the system, the current voltage, and the remaining capacity of the battery are captured. Second, the temperature of the battery, the motor drivers, and the motors are monitored with external temperature sensors. Third, the HC captures environment information, like environment temperature, air pressure, and humidity. Those external sensors are set up on a self-developed extension board for an Arduino Mega. The data of the HC is transmitted to the reflective operator. The reflective operator calculates the current condition of the system (the health) out of this data. If, for example, the battery is nearly empty but the current power consumption is high, the health of the system is in a critical condition. High temperatures on the motor drivers could either mean that the cooling of the system is not appropriate, or the motors are running too fast.

Also, as the HC has enough spare GPIOs, it connects a self-developed relay board with 10 relays which are used to switch components off or on. With the HC, it is possible to switch off the Kinect or any SBC of the system to reduce the power consumption.

A scheduler was developed to manage the flow of the multiple publishers, as Arduino's are usually not equipped with an OS.

### 4.3.4. Reflective Operator

The reflective operator of the DAEbot is located on another Raspberry Pi extended with a touch display to visualize and configure the robot and a CAN shield for its communication. The main task of the operator is to handle the information from the three underlying controllers. Therefore, state machines are developed with MATLAB Simulink (Stateflow) or Yakindu SCT. These state machines react to incoming CAN frames by executing predefined work sequences (e.g., stopping the motor if an obstacle is detected). It also generates the health status of the DAEbot by combining all of the data from the HC with the information from other modules of the distributed system, like data from the Internal Robot Controller. The current status of the robot is not healthy if, for example, the motors of the robot are not running but the temperature of the motor drivers is too high. The reflective operator defines in which mode the robot and all of its components are used. For example, if the battery is nearly empty, the robot switches to a power-saving mode to reduce energy consumption. This is done by setting the maximum velocity to a lower limit and by turning off the reflective operator+. The reflective operator is also used to process tasks like following an object. First, the reflective operator gets this task (e.g., with a display command executed by a human). Second, the reflective operator sets up the robots to fulfill this task. In this case, the motor driver and the CC need to be started. Third, the reflective operator processes the information from the CC (e.g., the distance to an object) and transmits new velocity values to the IRC.

The system's logic was designed model-based with state diagrams in several modules like battery management or motor logic. Independent services are designed to use the parallel computing capabilities of the multicore Raspberry Pi.

The reflective operator also (re)configures the controllers' publishers by setting up the publishers dynamically to other sample rates or priorities due to the current tasks and requirements.

Also, the operator can visualize data on the attached display. In particular, the data from the HC, like temperatures or environment information, can be monitored. The touch display is also used to change modes of the robot. It could, for example, be set to an energy-saving mode where the CC is entirely switched off.

### 4.3.5. Reflective Operator+

The goal of the reflective operator+ is to demonstrate the possibility to add computing resources to the operator whenever it is needed. The ZynqBerry TE0726-03M SBC has the form factor of a Raspberry Pi but uses a heterogeneous chip architecture consisting of a small FPGA SoC and an ARM-Cortex-A9-based co-microprocessor. Hence, the Operator+ can take over tasks from the reflective operator or even run massively parallel tasks (e.g., for computer vision tasks [36]) for which an FPGA is more suitable than the ARM-based Raspberry Pi. One parallel task which the reflective operator+ processes are OpenCV algorithms to detect obstacles in the video stream from the cameras. The reflective operator+ can be switched off with a relay when it is not used.

### 4.3.6. Cognitive Operator

The cognitive operator will run self-optimizing and planning algorithms as cloud services. It will extend the distributed system on the DAEbot with a remote cloud area. The cognitive operator will be connected via WiFi or mobile communications. Complex algorithms can be assigned to the cognitive operator, and all data from the robot will be saved in a database to run long-term analyses of the DAEbot.

### *4.4. Safety and Redundancy*

Each component of the DAEbot runs a local software watchdog to check if the connection to other components is working. The component reacts with a predefined strategy if the connection breaks down. For example, a loose connection of the IRC to the operators leads to stopping the motors of the mobile robot to prevent possible damage.

Self-healing behavior can be processed with the HC (e.g., by restarting the operator(s) if any dysfunction of it is detected). The operator then runs self-checking algorithms to solve its dysfunction (e.g., a CAN connection malfunction).

In other critical conditions, like high motor driver temperatures or a low battery, the HC transmits warnings to the reflective operator. These warnings are also visualized on the robot's display.

### 5. Results

The Operator–Controller Module (OCM) has successfully been adapted to robotics. The main result, the experiment to bring the conceptual model to a real-world robot, was successful. With the DAEbot, a distributed system within one robot has been developed. In contrast to systems with a single processing unit, architectures with a modular and distributed concept help to decouple their tasks and features on different levels. The conceptual model uses hard real-time constraints within the described controllers (e.g., to run the motors) and adds non-time-critical visualization on the operator layers. The usage of a distributed system also helps to integrate, for example, safety features or redundancies to a robot.

Some safety features, like terminating a communication channel, could successfully be tested with the DAEbot. With the implemented watchdog, the DAEbot automatically recognizes a communication loss (e.g., of the IRC), stops the robot immediately, and transmits a notification to the entire system.

The implemented communication framework fits into the architecture's main design characteristics with its ability to change the controller's publishing rate by the reflective operator. The described example, the adaptive publisher frequency (see Figure 4), allows the IRC to process sensor information only as fast as it is needed, which relieves the CPU usage. If, for example, the robot drives forward, the data from the ultrasonic sensor facing the back of the robot is not necessary. With the ability to set this publisher rate, the amount of frames on the network decreases, and with it, the possibility to have colliding CAN frames.

With its HC, the DAEbot can monitor the system and environment conditions. This condition information can be analyzed and used to adapt the distributed system to the robot's tasks in future projects. With the ability to monitor the system's energy consumption with the HC, strategies to save energy can be developed. The distributed system also allows saving energy by switching off sensors or processing units if they are not used. The DAEbot, for example, can activate a depth sensor with a relay if the results of the RGB cam are not sufficient.

The modular approach is necessary to provide the computing power using SBCs or small microcontrollers as processing units even for complex software algorithms. In the case of the DAEbot, widely used SBCs, like Arduino Mega, STM32 boards, or the Raspberry Pi, have been used. All of those SBCs fulfilled their tasks sufficiently. An FPGA board has been integrated to run massively parallel algorithms, like computer vision tasks. Bringing a robotic system into the cloud via the cognitive operator gives nearly unlimited resources to optimize the system.

The development process of the DAEbot was helpful to evaluate and refine the concept for the modular software architecture.

## 6. Conclusions

This article presents the concept of a modular system architecture for distributed robotic systems. This concept is based on the Operator–Controller Module (OCM) which was adapted for the usage of mobile robots. The article also outlines suggestions on communication or safety and redundancy.

With the evaluated design of the DAEbot, the presented architecture has successfully been applied. The DAEbot development outlines some aspects of its technical implementation. The experiment also shows that the decentralized approach in robots is useful, for example, to design scalable robot platforms which can easily be extended with other components.

In a future project, the DAEbot will be extended with an analysis tool for condition monitoring and fault-tolerance strategies. This analysis tool will monitor the current state of the robot with the reflective operator and long-term analysis of its conditions in the cloud. Therefore, the cognitive operator will be implemented with planning and self-optimizing algorithms. Cloud computing principles, especially for robotics [30,37] (e.g., neural network applications [29]), will be evaluated with the DAEbot. The DAEbot will also be used in a multirobot network [38] (e.g., alongside the AMiRo (Autonomous Mini Robot) [1]) to test the multilayer robot cooperation capabilities of the presented architecture. These future experiments will refine the proposed modular architecture for distributed robotic systems.

**Author Contributions:** Writing—original draft, U.J.; Writing—review & editing, C.W. and P.S.

**Funding:** This research received no external funding.

**Conflicts of Interest:** The authors declare no conflict of interest.

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
