# Peer review of "Concepts of a Modular System Architecture for Distributed Robotic Systems"

_computers, doi:10.3390/computers8010025_

Round 1

Reviewer 1 Report

I read this paper with much interest. It presents the concept of a modular system architecture for distributed robotic systems. This concept is based on the Operator-Controller Module which was adapted for the usage of mobile robots. The paper outlines suggestions on communication or safety and redundancy, which I find useful. I welcome any advance in this field. The presentation of this paper, however, can be improved. I will give my final recommendation based on the revision. The detailed comments are as follows.

(1) Line 12: is it better to add a word "frequently" in the sentence "Mobile robots are more and more designed in a modular manner..."?
(2) The contribution is only briefly mentioned in the last two sentences in Abstract. It is not a good balance. Please consider re-organize the abstract.
(3) The contribution of this paper should be further highlighted. In particular, the difference between this paper and the previous conference version should be clearly mentioned in Introduction.
(4) Related to the above comment, is it better to put Line 63 in a footnote?
(5) In line 73, what does "not described" mean here? Should it be deleted?
(6) In line 79-80, it is mentioned that the system mixes hard real-time and soft real-time. Does the system operate in both continuous-time and discrete-time (in the sense of 'switched systems')? For switched systems, please see the work 'Resilient consensus of switched multi-agent systems'. Please give a brief remark/clarification on this aspect.
(7) The conceptual model presented in Section 3 is interesting. It consists of three layers and hence can be modelled by mutliplex networks. A remark on this is recommended. See the seminal work Deffuant model of opinion formation in one-dimensional multiplex networks. This would link the current work to a wider audience.
(8) In Section 3.3, could you please explain more on time-discrete? I feel it enigmatic and not very clear about its meaning.
(9) The idea presented in lines 289-293 coincide with the resilience of complex networks. See the pertinent work 'Local natural connectivity in complex networks' on an essential metric of redundancy of general networks. It would be good to mention this idea so that the readers have a better understanding on its importance.
(10) Line 386 should either be expanded to explain Yocto project or combined with the previous paragraph.
(11) Section 5 is very short. However, there are about 10 paragraphs. Is is better to re-organize this section?
(12) The conclusion section can be expanded by adding some more discussions on open problems and future directions. This would be very instrumental in helping the interested readers of Computers.

Author Response

Thank you very much for your review. I added my comments in an attached .pdf file

Reviewer 2 Report

The paper presents a concept of a modular system architecture for distributed robotic systems. The architecture is based on the Operator-Controller Module (OCM), which splits a technical system hierarchically into a three-layer structure of controllers and operators.

An interesting paper with a promising application. The DAEbot is a nice tool for demonstration.

However, a central weakness of the paper is that it remains superficial in sections 4.3.3, 4.3.4. and 4.3.5 and that a discussion of the cognitive operator is lacking is section 4. Here the authors need to explain in much more detail the different tasks, processes, algorithms, … and need to give concrete examples.

Additionally, the topic of „health“ is not covered on the state of the art section – here a concise literature overview needs to be added and the different possibilities of health monitoring (and based on this of fault-tolerance)need to be discussed.

Generally the language is ok, but some mistakes are present e.g.:

line 265; by connection OF a new sensor to recognize objects to be future-proof

another language check is necessary.

All figures have some segments of letters in the lower right corner.

Author Response

(The authors gave the same response as above.)

Reviewer 3 Report

This paper proposes a modular system architecture for a robotic system and discusses the many benefits a modular architecture could provide such as safety, redundancy, integrating new sensors, etc.

Although the paper addresses an important area for robotic system and software development, there are a number of issues which needs to be addressed before publication. These issues are included below:

The contribution of the paper is not clear and many sections of the paper read like an applied design project report. It needs to be clarified what is it that authors are contributing to the state of art.

Section 3 is quite well-written but most of the discussion serves as a motivation and discusses the perceived benefits of modular architecture rather than explaining the conceptual model of the proposed approach.

Many acronyms are not defined: E.g., DDX, IDEs, SERA

Some claims are not substantiated by references or justified in the case of missing references: E.g., line 46 – “Often distributed systems grow historically” and line 77 – “accepted as standard framework for robotic system development”

The terminology also needs to clarified. What is a definition of distributed systems. In many fields, such as system engineering, a distributed system means where systems components are physically or geographically distributed. It appears that authors are using distributed system as a synonym for modular system. If that is the case then this should be clearly specified in the paper. E.g. line 16 “modular system architecture was designed for distributed robotics systems” implies that robots are distributed which is not the case in this paper.

The paper needs to be revised for clarity in writing, especially abstract and introduction. There are a few fragment sentences and typographical error in the paper. For example:

Line 57 – ‘if’

Line 99 “is done and an adaptive layer’

System Architecture is a well established field of research in Systems Engineering and there are number of papers on examining/quantifying/measuring modularity of a system architectures.

 For Example on System Architecture, see:

Crawley E, Cameron B, Selva D. System Architecture: Strategy and Product Development for Complex Systems. 1st ed. Boston: Pearson; 2015.

Raz, Ali K., C. Robert Kenley, and Daniel A. DeLaurentis. "System architecting and design space characterization." Systems Engineering 21.3 (2018): 227-242.

For modularity of System Architecture:

Refer to Design Structure Matrix and/or N-squared diagrams and the following reference

Raz, Ali K., and Daniel A. DeLaurentis. "System-of-Systems Architecture Metrics for Information Fusion: A Network Theoretic Formulation." AIAA Information Systems-AIAA [email protected] Aerospace. 2017. 1292.

Examining and comparing how your approach to modularity adds value compared to a central design could be very valuable contribution.

The results section is a summary of observations. Details of what experiments were performed to substantiate the results are largely missing.

Author Response

(The authors gave the same response as above.)

Round 2

Reviewer 1 Report

The authors have made a thorough revision according to me comments in the first round of review. The paper has been improved. Therefore, I would like to recommend it for publication in Computers. 

Author Response

Thank you for your second review. Please check the attached Notes.

Reviewer 2 Report

Thank you for the improvement, but I am afraid that the problem with the new section 2.2 as well as 4.3.3, 4.3.4, 4.3.6 as well as the lack of a description of the cognitive operator remains.

The literature chosen in 2.2 seems arbitrary. In 4.3.3 you claim to capture several types of information but the aggregation to some kind of health information remains unclear. The logic and algorithms in the reflective operator remain unclear.

I am sure that a more concrete explanation is possible without losing the main focus of the paper. It is not important to show ALL algorithms but to show some in detail in order to be able to understand the interplay in the robot.

Author Response

(The authors gave the same response as above.)

Reviewer 3 Report

The paper is much improved and easier to read. Following are some additional suggestions:

Line 26: DAEbot is mentioned in abstract. It will be helpful to define what is a DAEbot. A novice may not have an idea about it.

Line 96: Double check the formatting for Reference 11 in references. It appears as blank

Line 449 – 450 the Cam controller instead of cam

Line 444 – It is better to not start a sentence with E.g., you spell out as For example,

Line 452…”logic was design model-based” is somewhat incomplete phrase. Can you specify the model-based use here? Perhaps maybe include the model of the logic that was designed.

Line 499 – What system are the notification sent to. This is a little bit confusing and can be clarified.

Line 502 – Spelling ‘descibed’

Line 502 - 506: Please re-examine these sentences. They read as the ability to set publisher rates is an undesired outcome. Is that correct?

Line 541 – and Line 45: specify in parenthesis what is AMiRo

Line 542: suggest replacing those with ‘these future experiments ….”

Reviewer Comments to Authors responses to round 1 review:

 -> For “often distributed system grow historically”: You’re right. A reference for this is missing. I searched for a reference for this a lot but didn’t find one. I know from my experience and my colleagues that this happens a lot in the field of robotics, but “bad” results are typically not published. Do you have a suggestion on how to solve it?

 Refer to a reference from Mark Maier on Architecting Principles of System-of-Systems. This talks about the idea of evolutionary development of distributed which means that not elements of a distributed systems are are developed/upgraded/evolved at the same time.

Maier, Mark W. "Architecting principles for systemsofsystems." Systems Engineering: The Journal of the International Council on Systems Engineering 1.4 (1998): 267-284.

-> Thanks for these interesting recommendations. I will order the book as soon as possible but I’m afraid that it is to late add it for this article. I cited some architectures and compared them slightly to our approach in SOA (references 11 to 19)

Understand ordering a book can take time and resources. Suggest using the above Mark Maier reference with Raz et. al. System Architecting reference (listed in round 1 review with book name) to provide a decent coverage.

Author Response

(The authors gave the same response as above.)

Round 3

Reviewer 2 Report

I have hoped for a bit stronger improvement, but I think in tis form the article can be accepted.

Computers EISSN 2073-431X Published by MDPI AG, Basel, Switzerland RSS E-Mail Table of Contents Alert
Back to Top